# Molecular phyloecology suggests a trophic shift concurrent with the evolution of the first birds

Yonghua Wu [1,2]✉

Birds are characterized by evolutionary specializations of both locomotion (e.g., flapping flight) and digestive system (toothless, crop, and gizzard), while the potential selection pressures responsible for these evolutionary specializations remain unclear. Here we used a recently developed molecular phyloecological method to reconstruct the diets of the ancestral archosaur and of the common ancestor of living birds (CALB). Our results suggest a trophic shift from carnivory to herbivory (fruit, seed, and/or nut eater) at the archosaur-to-bird transition. The evolutionary shift of the CALB to herbivory may have essentially made them become a low-level consumer and, consequently, subject to relatively high predation risk from potential predators such as gliding non-avian maniraptorans, from which birds descended. Under the relatively high predation pressure, ancestral birds with gliding capability may have then evolved not only flapping flight as a possible anti-predator strategy against gliding predatory non-avian maniraptorans but also the specialized digestive system as an evolutionary tradeoff of maximizing foraging efficiency and minimizing predation risk. Our results suggest that the powered flight and specialized digestive system of birds may have evolved as a result of their tropic shift-associated predation pressure.

[1] School of Life Sciences, Northeast Normal University, Changchun, China. [2] Jilin Provincial Key Laboratory of Animal Resource Conservation and Utilization, Northeast Normal University, Changchun, China. ✉email: wuyh442@nenu.edu.cn

Diet plays a fundamental role in the life of an animal. It defines interactions with other organisms and shapes their evolution. Modern birds exhibit diverse diet preferences, including herbivory, omnivory, and carnivory, whereas the diet of ancestral birds remains less clear. Fossil evidence shows that ever since the origin of birds (Aves: defined herein as the clade including *Archaeopteryx* and modern birds, as proposed previously[1–3]) from the Late Jurassic, they had undergone adaptive radiation to diversified dietary niches in the Cretaceous, with herbivorous (e.g., fruits and seeds), piscivorous, and insectivorous diets found[3–10]. In particular, seed and/or fruit eating are suggested in many ancestral bird lineages, such as *Jeholornis*, *Confuciusornis*, *Sapeornis*, *Hongshanornis*, and *Yanornis*[2–8,10–13]. This suggests that seed and/or fruit eating may have been relatively common during the early evolution of birds, and that this herbivorous adaptation may play a vital role in the early evolution of birds[5,11,14,15].

Studies in comparative digestive physiology provide important insights into understanding the molecular bases underlying the dietary variation of animals[16,17]. Accumulating evidence has revealed a fundamental pattern that the digestive physiology of animals evolves in parallel with their diets[16–18]. Further, the digestion and absorption capability of animals generally reflects their dietary load of different nutrient substrates such as carbohydrates, proteins, and fats[17,19–22]. The higher the nutrient substrate, the higher the expression and activity of its corresponding digestive enzymes and nutrient transporters, and vice versa[16,17]. This suggests that the digestion and absorption capability of animals is under evolutionary adaptation to approximately match loads of different dietary components such as carbohydrates, proteins, and fats in their diets[16,17,23–26]. With this in mind, one would expect that herbivores and carnivores may tend to present an evolutionary enhancement of the digestion and absorption of plant food and meat, respectively. Regarding plant food and meat (including both invertebrates and vertebrates), one of the important differences between them is that meat is generally high in proteins and fats, whereas plant food is generally high in carbohydrates[17,23,25,27,28], except seeds, in particular nuts, which are rich in fat as well[27]. Indeed, recent studies on the evolution of digestive system-related genes have shown that carnivores more likely show an evolutionary enhancement of the genes related to the digestion and absorption of proteins and fats, whereas animals consuming abundant plant foods (e.g., herbivores and omnivores) tend to exhibit an evolutionary enhancement of the genes related to the digestion and absorption of carbohydrates[25,28,29], with the exception of parrots, which ingest seeds and nuts, and present an evolutionary enhancement of the digestion and absorption of fats in addition to carbohydrates[29]. This may suggest that the adaptive evolution of digestive system-related genes is capable of reflecting the dietary variations of different animals[25,28,29].

The recent development of a molecular phyloecological approach, which employs the phylogenetic evolutionary analyses of the molecular markers indicative of trait states, allows us to reconstruct ancestral traits using molecular data[30–32]. The substantial dietary differences between carnivores and herbivores in terms of the amounts of dietary components (e.g., carbohydrates, proteins, and fats)[17,23,25,27], and the adaptation of digestive system-related genes to the variations of dietary components of animals[16,17,23–26,29] may suggest that digestive system-related genes can be used as the molecular markers of diets to reconstruct the diets of ancestral animals in the context of molecular phyloecology[29]. In this study, we employed the molecular phyloecological approach using digestive system-related genes as molecular markers to infer the diets of the ancestral archosaur and of the common ancestor of living birds (CALB). Our results revealed a diet shift from carnivory to herbivory at the archosaur-to-bird transition. The molecular findings of the diet shift, coupled with the research advance of avian paleontology, provide new insights into understanding the origin of birds.

## Results

We examined the adaptive evolution of 83 digestive system-related genes (Supplementary Data 1) in the context of sauropsid phylogeny (Fig. 1). The 83 genes came from three digestive system-related Kyoto Encyclopedia of Genes and Genomes (KEGG) pathways including carbohydrate digestion and absorption (CDA), protein digestion and absorption (PDA), and fat digestion and absorption (FDA) (Fig. 2). The functions of these digestive system-related genes are relatively well-studied and are known to play important roles in the digestion and absorption of carbohydrates, proteins, and fats. Following the molecular phyloecological approach to reconstruct ancestral traits[30–32], we used branch and branch-site models implemented in PAML software[33] to detect positively selected genes (PSGs) along our target branches, and PSGs were found based on branch-site model (Table 1). The evidence of the positive selection of genes may suggest a functional enhancement of their corresponding functions in relevant lineages[30–32].

We initially analyzed the positive selection of the digestive system-related genes along the common ancestor branch of living birds. Among the 83 genes analyzed, we found 17 PSGs across all three pathways, with CDA and FDA showing relatively strong positive selection and PDA showing the relatively weakest positive selection in terms of *p*-values and the number of PSGs (Table 1 and Fig. 2). For CDA, seven PSGs (*ATP1B3*, *ATP1B4*, *HK3*, *SLC5A1*, *LCT*, *SI*, and *SLC2A5*) were found with generally low *p*-values compared to those PSGs found in FDA and PDA. Among the seven PSGs found, three genes, *SI*, *LCT*, and *HK3*, showed relatively strong positive selection signals. *SI* encodes sucrase-isomerase and is essential for the digestion of dietary carbohydrates, such as starch, sucrose, and isomaltose[34]. *LCT* shows lactase activity and phlorizin hydrolase activity[35]. *HK3* is involved in glucose metabolism[36]. Similar to *HK3*, one PSG, *SLC5A1*, also plays a role in glucose metabolism, functioning as a transporter of glucose in the small intestine[37]. Intriguingly, we detected the positive selection of one gene, *SLC2A5*, encoding GLUT5, which is known to have an exclusive affinity for fructose (fruit sugar) and is the major fructose transporter in the intestines and other tissues, mediating the uptake of dietary fructose[38–40]. We also detected the positive selection of *ATP1B3* and *ATP1B4*, which encode the Na+/K+ ATPase involved in the CDA pathway to maintain ionic homeostasis[41]. Besides CDA, we detected PSGs involved in the FDA pathway and eight PSGs (*ABCG5*, *AGPAT1*, *AGPAT2*, *APOA1*, *APOA4*, *APOB*, *CD36*, and *NPC1L1*) were found. Of these, *CD36* plays an important role in the uptake and processing of fatty acids[42]. *NPC1L1* is involved in the intestinal absorption of cholesterol and plant sterols[43]. *AGPAT1* and *AGPAT2* play roles in converting lysophosphatidic acid into phosphatidic acid[44]. *APOA1*, *APOA4*, and *APOB* encode key apolipoproteins to carry fats and fat-like substances in the blood[45,46]. Remarkably, we found the positive selection of one gene, *ABCG5*, which encodes sterolin-1 and works together with sterolin-2, encoded by gene *ABCG8*, to form a protein called sterolin. Sterolin is a transporter protein and plays an important role in eliminating plant sterols to regulate the whole-body retention of plant sterols[43,47], which are mainly present in nuts and seeds[43]. Unlike CDA and FDA, relatively weak positive selection signals were found in PDA with only two PSGs (*CELA3B* and *SLC36A1*) clearly involved in protein utilization[48,49], whereas the other two PSGs (*ATP1B3* and

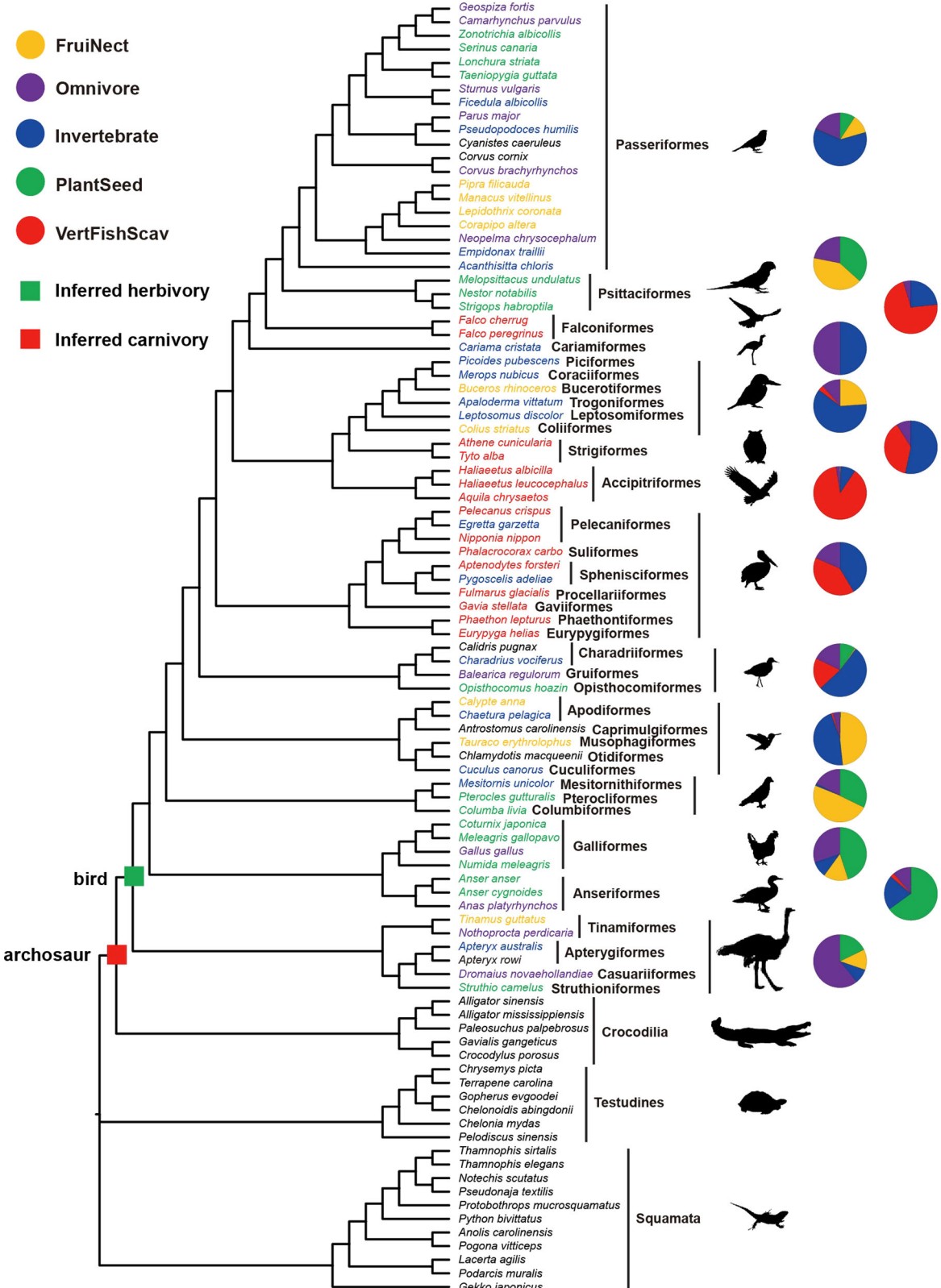

**Fig. 1 Phylogeny and the diets of modern birds.** Phylogenetic relationships of taxa used follow published studies[30,138–143]. Dietary categories of each bird species follow one published study[145] and are shown in different colors, and the bird species without dietary information available are shown in black. The dietary categories of avian clades based on the dietary data of a total of 9993 extant bird species are shown in pet charts. PlantSeed (plant and seeds), FruiNect (fruits and nectar), Invertebrate (invertebrates), VertFishScav (vertebrates and fish and carrion), and Omnivore (score of ≤50 in all four categories).

**a** Carbohydrate digestion and absorption

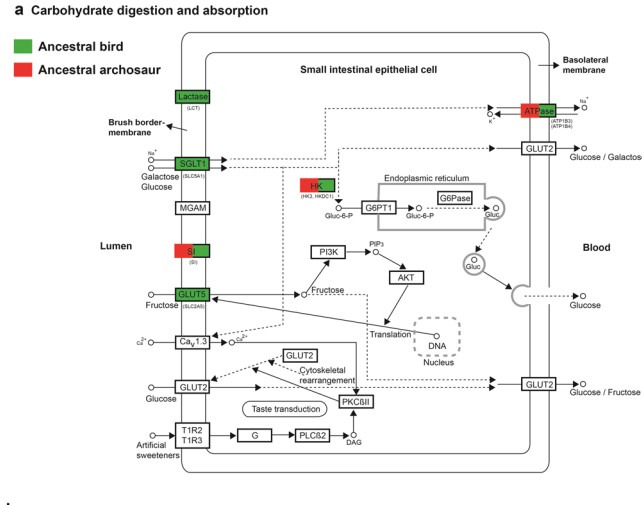

**b** Protein digestion and absorption

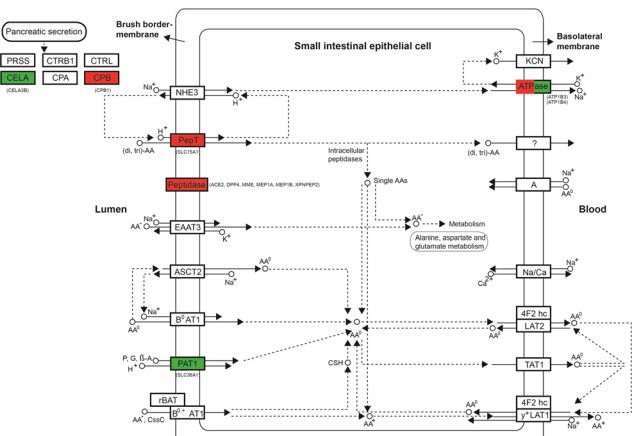

**c** Fat digestion and absorption

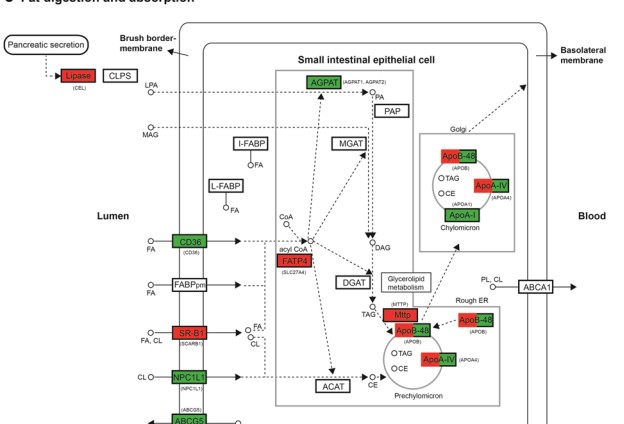

**Fig. 2 Digestive system pathways and positively selected genes.** The digestion and absorption of carbohydrates (**a**), proteins (**b**), and fats (**c**) are shown. The proteins with their corresponding genes in parentheses under positive selection are highlighted in red (ancestral archosaur) and green (ancestral bird). The three digestive system pathways are modified from corresponding KEGG pathways with accession numbers (map04973, map04974, and map04975).

ATP1B4), which are involved in both CDA and PDA, play a role in maintaining ionic homeostasis (Fig. 2).

We subsequently examined the positive selection of the digestive system-related genes along the common ancestor

branch of living birds and crocodilians, representing the ancestral archosaur. In contrast to the CALB, for the ancestral archosaur, we detected the highest number of PSGs in PDA, followed by FDA, with the lowest number of PSGs found in CDA (Fig. 2 and Table 1). For PDA, eight PSGs were found, of which seven PSGs (*ACE2, CPB1, DPP4, MEP1A, MEP1B, MME,* and *XPNPEP2*) encode peptidases[50–55] and one gene, *SLC15A1*, is involved in the intestinal transport of peptide[56]. For FDA, we found six PSGs, including *APOA4, APOB, CEL, MTTP, SCARB1,* and *SLC27A4*, of which *CEL* encodes a bile salt-dependent carboxyl-ester lipase, hydrolyzing dietary fats, and cholesteryl esters in the small intestine[57]. *SCARB1* mediates the uptake of cholesterol and lipids[58]. *SLC27A4* is an important fatty acid transporter in small intestinal enterocytes[59]. *MTTP* is involved in the transport of triglycerides, cholesteryl esters, and phospholipids[60]. *APOA4* and *APOB* encode two key apolipoproteins responsive to carrying fats and fat-like substances in the blood[45,46]. Unlike PDA and FDA, CDA showed the lowest number of PSGs and only three PSGs (*SI, HKDC1,* and *ATP1B4*) were detected. *HKDC1* is involved in glucose metabolism[61]. *ATP1B4* encodes $Na^+/K^+$ ATPase to maintain ionic homeostasis[41], involved in both CDA and PDA (Fig. 2). *SI* encodes sucrase-isomerase, digesting dietary carbohydrates including starch, sucrose, and isomaltose[34].

Our positive selection analyses described above showed that the CALB exhibited a predominant Darwinian selection of the genes related to CDA and FDA, whereas the ancestral archosaur exhibited a marked Darwinian selection of the genes related to PDA and FDA, indicating substantial selection differences between them (Fig. 2 and Table 1). To further know the possible selection differences between the CALB and the ancestral archosaur, we then used the program RELAX[62] to examine the selection intensity changes of the digestive system-related genes of the CALB relative to those of the ancestral archosaur (Supplementary Data 2). Our results showed that FDA-related genes exhibited the most intensified selection, followed by CDA-related genes, whereas PDA-related genes showed the relatively weakest selection intensification. For FDA, eight genes (*ABCA1, AGPAT1, CD36, FABP1, MTTP, NPC1L1, GOT2,* and *SCARB1*) exhibited a relatively intensified selection. Among the eight genes, five (*AGPAT1, CD36, MTTP, NPC1L1,* and *SCARB1*), as mentioned above, are mainly related to the transport or conversion of lipids, whereas the other three genes (*ABCA1, FABP1,* and *GOT2*) are mainly involved in the transport of lipids. Specifically, *ABCA1* mediates cellular cholesterol and phospholipid efflux[63], *FABP1* regulates lipid transport and metabolism[64], and *GOT2* is involved in fatty acid transport[65]. Besides FDA, three CDA-related genes, *PIK3CB, SLC37A4,* and *CACNA1D*, showed selection intensification as well. *PIK3CB* encodes the catalytic subunit of phosphoinositide 3-kinase, which plays a role in regulating the activity of GLUT5, a major fructose transporter[66]. *SLC37A4* acts as a transporter of glucose 6-phosphate[67]. *CACNA1D* encodes a subunit of a calcium channel (CaV1.3) (Fig. 2). For PDA, only one gene, *CELA3B*, which encodes pancreatic serine proteinases[49], was subject to selection intensification. In addition to these selection-intensified genes, several genes showed relative selection relaxation in the CALB compared to the ancestral archosaur, including two genes (*ABCG5* and *APOB*) of FDA, one gene (*CTRL*) of PDA, and one gene (*SLC5A1*) clearly involved in CDA.

## Discussion

Comparative digestive physiology studies demonstrate that the evolution of digestive system molecules adapts for the amounts of nutrient components (e.g., carbohydrates, fats, and proteins) in the diets of animals[16,17]. Our results showed that the ancestral archosaur exhibited a marked selection of the genes related to

**Table 1 Positively selected genes of ancestral bird and ancestral archosaur identified by branch-site model.**

| Taxa/gene | N | 2ΔlnL | df | P-value | ω | Positively selected sites |
|---|---|---|---|---|---|---|
| **Ancestral bird** | | | | | | |
| ABCG5 | 75 | 5.32 | 1 | 0.021 (0.042) | $\omega_{2a} = 998.99$ $\omega_{2b} = 998.99$ | 135S, 290T, 483S |
| AGPAT1 | 40 | 74.72 | 1 | 5.424E − 18 (1.084E − 17) | $\omega_{2a} = 155.38$ $\omega_{2b} = 155.38$ | 14I, **183R** |
| AGPAT2 | 90 | 4.02 | 1 | 0.045 (0.090) | $\omega_{2a} = 14.66$ $\omega_{2b} = 14.66$ | **31C**, 116M |
| APOA1 | 50 | 5.86 | 1 | 0.015 (0.031) | $\omega_{2a} = 998.99$ $\omega_{2b} = 998.99$ | **126Q** |
| APOA4 | 41 | 9.06 | 1 | 0.003 (0.005) | $\omega_{2a} = 334.64$ $\omega_{2b} = 334.64$ | 25D, 49A, 64D, 79K, **92V**, 97K, 126R, 133A, 144T, 156K, 158E, **162Q**, 180S, 224S, 233Q, 247K, 255Q, **270L**, 277K, 288S, 295G, 319N, 362L, 365I |
| APOB | 69 | 11.99 | 1 | 0.001 (0.001) | $\omega_{2a} = 10.51$ $\omega_{2b} = 10.51$ | 84S, 248A, 981E, 1704A, 1926Q, 3722N, 4514Y |
| ATP1B3 | 83 | 10.56 | 1 | 0.001 (0.002) | $\omega_{2a} = 37.99$ $\omega_{2b} = 37.99$ | **60N**, 95L, **120L** |
| ATP1B4 | 84 | 6.36 | 1 | 0.012 (0.023) | $\omega_{2a} = 998.99$ $\omega_{2b} = 998.99$ | 22K, 29A, **30K**, 103I, 135K, **188T** |
| CD36 | 77 | 4.58 | 1 | 0.032 (0.065) | $\omega_{2a} = 29.60$ $\omega_{2b} = 29.60$ | 75M, 246N, **250I**, 292E, 371S |
| CELA3B | 15 | 4.98 | 1 | 0.026 (0.026) | $\omega_{2a} = 998.93$ $\omega_{2b} = 998.93$ | 87D, **182I**, 199A |
| HK3 | 48 | 14.52 | 1 | 0.000 (0.000) | $\omega_{2a} = 520.78$ $\omega_{2b} = 520.78$ | 57I, 61Q, 88Q, **115C**, 296D, 362P, **522A**, 528E, 609I, 726R |
| LCT | 71 | 15.20 | 1 | 9.670E − 05 (1.934E − 04) | $\omega_{2a} = 999.00$ $\omega_{2b} = 999.00$ | 251Q, 1790T, 1810S |
| NPC1L1 | 34 | 6.06 | 1 | 0.014 (0.028) | $\omega_{2a} = 697.54$ $\omega_{2b} = 697.54$ | 3S, 10L, 12F, 17F, 53Y, 83V, 93S, **94S**, 95W, 107G, 112L, 120S, **125Y**, 127Y, 157L |
| SI | 46 | 25.78 | 1 | 3.826E − 07 (7.653E − 07) | $\omega_{2a} = 218.16$ $\omega_{2b} = 218.16$ | 69V, 178V, 309-, 347-, 382L, 923A, **1057G**, **1204Y**, **1291W**, 1292G, **1297Y**, 1432Y, **1453S**, **1535F**, 1663-, **1707H** |
| SLC2A5 | 28 | 10.16 | 1 | 0.001 (0.001) | $\omega_{2a} = 18.45$ $\omega_{2b} = 18.45$ | 85L, 93F, **108S**, 222E, 250A, **264S**, 342-, 355A, **411S**, 440A, 459G |
| SLC36A1 | 15 | 10.00 | 1 | 0.002 (0.002) | $\omega_{2a} = 222.64$ $\omega_{2b} = 222.64$ | 20S, 112P, **171E**, 212T, **215R**, 308T, 334S, **430S** |
| SLC5A1 | 67 | 3.92 | 1 | 0.048 (0.095) | $\omega_{2a} = 19.29$ $\omega_{2b} = 19.29$ | **470S** |
| **Ancestral archosaur** | | | | | | |
| ACE2 | 76 | 5.06 | 1 | 0.024 (0.049) | $\omega_{2a} = 53.65$ $\omega_{2b} = 53.65$ | 114Q, 142P, **181S**, 641Y, 699E, 712N, 797T |
| APOA4 | 41 | 4.92 | 1 | 0.027 (0.053) | $\omega_{2a} = 44.66$ $\omega_{2b} = 44.66$ | 192Q, 265L, 310V, 339E |
| APOB | 69 | 7.76 | 1 | 0.005 (0.011) | $\omega_{2a} = 13.97$ $\omega_{2b} = 13.97$ | 401A, 1151D, **171N**, 1625E, **1687D**, 1791Y, 2354K, 4096H |
| ATP1B4 | 84 | 14.16 | 1 | 0.000 (0.000) | $\omega_{2a} = 139.31$ $\omega_{2b} = 139.31$ | 6A, **43Q**, 45M, 153D, **186Y**, 193S |
| CEL | 66 | 14.64 | 1 | 0.000 (0.000) | $\omega_{2a} = 151.47$ $\omega_{2b} = 151.47$ | 5-, 11-, 97E, 154K, 293V, 345G, 383Y |
| CPB1 | 51 | 10.62 | 1 | 0.001 (0.002) | $\omega_{2a} = 998.99$ $\omega_{2b} = 998.99$ | 142S, **143K**, 195R, 293N, 409I |
| DPP4 | 86 | 8.26 | 1 | 0.004 (0.008) | $\omega_{2a} = 39.05$ $\omega_{2b} = 39.05$ | 76A, 166S, 286L, **293F** |
| HKDC1 | 86 | 5.56 | 1 | 0.018 (0.037) | $\omega_{2a} = 47.15$ $\omega_{2b} = 47.15$ | 434V, 474H, **820N** |

**Table 1 (continued)**

| Taxa/gene | N | 2ΔlnL | df | P-value | ω | Positively selected sites |
|---|---|---|---|---|---|---|
| MEP1A | 71 | 3.92 | 1 | 0.048 (0.095) | $\omega_{2a} = 30.69$ $\omega_{2b} = 30.69$ | 229*, **311I**, 373Q, 644- |
| MEP1B | 84 | 6.20 | 1 | 0.013 (0.026) | $\omega_{2a} = 111.71$ $\omega_{2b} = 111.71$ | 346T |
| MME | 90 | 7.46 | 1 | 0.006 (0.013) | $\omega_{2a} = 540.88$ $\omega_{2b} = 540.88$ | 94K, 375Y, **446Q**, 499D |
| MTTP | 89 | 4.38 | 1 | 0.036 (0.073) | $\omega_{2a} = 26.96$ $\omega_{2b} = 26.96$ | 28I, 43L, 49G, 51S, 127W, 183N, 207L, 548F, 578L, 784V, **889E** |
| SCARB1 | 90 | 5.54 | 1 | 0.019 (0.037) | $\omega_{2a} = 210.06$ $\omega_{2b} = 210.06$ | 206S |
| SI | 46 | 7.78 | 1 | 0.005 (0.011) | $\omega_{2a} = 64.88$ $\omega_{2b} = 64.88$ | 66E, 208V, 335-, **398S**, 607V, 650S, 1342Y |
| SLC15A1 | 76 | 21.38 | 1 | 3.767E − 06 (7.534E − 06) | $\omega_{2a} = 613.86$ $\omega_{2b} = 613.86$ | 320H, 484T, 514Y, 568A |
| SLC27A4 | 85 | 6.06 | 1 | 0.014 (0.028) | $\omega_{2a} = 22.11$ $\omega_{2b} = 22.11$ | 43R, 153A, 170G, **268S**, 349K, **397K**, 433Q, 475E, 563L |
| XPNPEP2 | 65 | 9.90 | 1 | 0.002 (0.003) | $\omega_{2a} = 198.45$ $\omega_{2b} = 198.45$ | 30I, **66Q**, **144D** |

$\omega$ values for foreground branch are shown. P-value under Bonferroni multiple testing correction is shown in parenthesis. Positively selected sites with ≥90% probability support are shown in bold. N represents the species number used for the positive selection analyses of each gene.

PDA and FDA, whereas the CALB presented a predominant selection of the genes involved in CDA and FDA (Fig. 2, Table 1, and Supplementary Data 2). These results remained largely unchanged even after the Bonferroni multiple testing correction of the *p*-values of PSGs (Table 1). Especially for the ancestral archosaur, our positive selection analyses revealed the highest number of PSGs in PDA, followed by FDA, with the lowest number of PSGs found in CDA (Fig. 2). This may suggest that the diet of the ancestral archosaur was characterized by a high amount of proteins, followed by fats, with a minimum load of carbohydrates. This nutrient profile is highly consistent with the presumable carnivory of ancestral archosaurs[68], as meats are generally rich in proteins, followed by fats, with the minimum amount of carbohydrates[27]. Contrary to the ancestral archosaur, for the CALB, our results based on two different methods (PAML and RELAX) consistently demonstrated that it showed a relatively strong selection in CDA- and FDA-related genes, with the weakest selection found in PDA-related genes (Fig. 2, Table 1, and Supplementary Data 2). This may suggest that the diet of the CALB is characterized by high amounts of carbohydrates and fats, with a relatively minimal amount of proteins, representing a high-energy diet. This seems to be more consistent with herbivory compared to carnivory, considering that plant foods are rich in carbohydrates, whereas meats are particularly high in proteins[27]. In particular, most PSGs involved in CDA (*SI*, *SLC5A1*, *SLC2A5*, *HK3*, and *LCT*) were found in the CALB center on the digestion and absorption of sugars (e.g., glucose, sucrose, and fruit sugar), indicating its high-sugar diet. A high-sugar diet may suggest their eating of fruits, which are characterized by relatively high amounts of sugars among plant foods[24,27,69]. In particular, one PSG, *SLC2A5*, found in the CALB is mainly involved in the transports of fruit sugar[38–40]. These lines of evidence may suggest that the CALB involved fruits in its diet. On the other hand, for the PSGs found in FDA, one gene, *ABCG5*, plays a critical role in the transport of plant sterols, which are mainly found in nuts and seeds[43,47]. This may suggest that the CALB ingests seeds and/or nuts as well, which are rich in fat[27]. The predominant selection of the CALB in FDA is similar to parrots, which consume considerable amounts of seeds and nuts, and are found to present a strong Darwinian selection in FDA as well with four PSGs found, of which three (*ABCG5*, *APOA4*, and *APOB*) are shared with the CALB[29]. In all, our molecular study suggests that the ancestral archosaur is probably a carnivore, whereas the CALB is more likely an herbivore, ingesting fruits, seeds, and/or nuts (Fig. 1).

Regarding digestive system-related genes, in addition to diets, other factors (e.g., flight and microbial fermentation) may affect their evolution as well. With respect to flight, previous studies show that in favor of flight, fliers (e.g., birds and bats) have evolved to have a smaller intestinal size and shorter retention times of digesta relative to nonfliers[70–72], and thus there may be an increased selection for the digestion and absorption of nutrient substrates as a compensation for the constraints on the digestive system in fliers[70,72]. This may alternatively explain our observed enhanced selection of the digestion and absorption of carbohydrates and fats of the CALB; however, it is difficult to interpret why such an enhanced selection was not found in the PDA of the CALB, as observed in this study (Table 1). Moreover, previous studies show that the increased digestion and absorption of nutrient substrates in fliers (birds) compared to nonfliers (mammals) seem to be restricted to the paracellular absorption pathway, in which nutritional substances move through the tight junctions adjoining cells, rather than the transcellular absorption pathway, which include CDA, PDA, and FDA, as examined in this study[70,72]. Thus, the possible effects of flight on the evolution of the digestive system-related genes of the CALB examined in this study may be relatively small. In addition to flight, microbial

fermentation, which transfers dietary carbohydrates (e.g., cellulose) to volatile fatty acids and microbe proteins for the utilization of herbivores, may be another possible factor that affected the evolution of digestive system-related genes; however, its importance is considered to be mainly restricted to herbivores (e.g., ungulates) that rely mainly on microbial fermentation and is relatively trivial to other animals[16]. These lines of evidence may suggest that the selection differences of the digestive system-related genes observed in the CALB and the ancestral archosaurs may be mainly due to their dietary differences, although there exist possible effects of flight and microbial fermentation on the evolution of their digestive system-related genes.

Our molecular results are highly consistent with the fossil evidence showing that ancestral archosaurs are generally typically meat eaters[68] and a great number of ancestral Mesozoic birds, including the basal birds, such as *Jeholornis*, *Confuciusornis*, and *Sapeornis*, show features or gut contents indicating that they ate fruits and/or seeds[2,3,5–8,10,12]. In particular, for the herbivory of the CALB found in this study, it is consistent with the widespread herbivory observed in many living bird lineages across bird phylogeny (Fig. 1). In line with this, one previous study shows evidence of seeds as an important dietary component of the CALB using maximum likelihood reconstruction[73]. Considering that ancestral birds lived in a conifer-dominated ecosystem[9,74], the seeds that they ate might partly come from conifers[11]. Indeed, the seeds of many conifers (e.g., pines) are relatively rich in lipids[75,76], which might have led to the evolutionary enhancement of FDA of the CALB found in this study. In addition, previous studies show that the Late Jurassic/Early Cretaceous radiation of more advanced birds temporally coincides with that of angiosperm plants[77] and it is likely that the fruit- and/or seed-eating habitat of ancestral birds may have partly helped for their dispersal of seeds[2]. The herbivory of the CALB is also consistent with the occurrence of ceca observed in the majority of living birds, including the basal lineages (e.g., ratites), which is generally considered to be helpful for cellulose digestion and fermentation linked to herbivory[18,78]. The dietary shift of the CALB to herbivory is also consistent with the observation of reductions in both the teeth[3,8] and biting force[79,80] across the theropod-bird transition, which is considered to have resulted from the dietary shift from carnivorous to herbivorous diets[15,79]. The similar transition from carnivory to herbivory occurs multiple times in theropods[15,81,82]. The causes underlying the evolutionary shift to the herbivory of the CALB are not clear, but the possible competition from carnivorous theropods and pterosaurs is proposed as a possible candidate[14,79]. The finding of the herbivory of the CALB ingesting fruits, seeds, and/or nuts, which characterize seed plants adapted to dry land environments[83], may strongly suggest that the CALB mainly occurred in terrestrial habitats rather than an aquatic environment, as hypothesized previously[84]. These findings are consistent with the fact that the phylogenetically most basal extant neornithine birds—i.e., Palaeognathae and Galloanseres—are predominantly herbivorous or omnivorous and they mainly occur in terrestrial habitats[2].

Our results demonstrate an evolutionary shift of the CALB to an herbivorous diet (fruit, seed, and/or nut eater) (Fig. 1), suggesting that the CALB may be a low-level consumer. Evolutionarily, birds are widely believed to be derived from a group of small maniraptoran theropods, including dromaeosaurids and troodontids[2,4,5,85]. Among these maniraptoran theropods, many of them, including most dromaeosaurids and derived troodontids, show carnivory[2,3,5,8,15,82,86–91] (Fig. 3). However, unlike their maniraptoran relatives, many bird lineages, including the basal bird lineages, such as *Jeholornis* and *Sapeornis*, may have evolved to exploit herbivorous niches, as evidenced by both the molecular

(Fig. 1) and fossil evidence mentioned above[2–8,10,11,15,79] (Fig. 3). The dietary shift from carnivory to herbivory may suggest a shift of the trophic niche of bird ancestors from that of a high-level consumer to a low-level consumer as a primary and/or secondary consumer[74,89]. This is consistent with the marked reduction or loss of teeth along with the evolution of birds[3–5,92], a feature indicative of low-level consumers rather than high-level consumers (e.g., top predators), which would otherwise show a predation feature of well-developed teeth[86,93]. Moreover, although diverse diets (e.g., seeds, fish, and insects) among ancestral bird lineages have been found, there is no direct fossil evidence indicative of their preying on terrestrial vertebrates[3], strengthening their ecological niches as low-level consumers. Ancestral birds were abundant in Mesozoic terrestrial ecosystems[74], occurring globally[94] and representing a potential food source for carnivores. Becoming a low-level consumer, ancestral birds may be under increased predation risk. This is particularly the case for ancestral birds, as they evolve toward miniaturization suitable for powered flight[95,96] and their small body size may be particularly vulnerable to predators. More importantly, their evolution of endothermy and powered flight requires much more energy and, consequently, frequent foraging[3,5,30,97]. Frequent foraging may have, most often, exposed them to predators, hence leading to their high predation risk. In support of this, fossil evidence shows that ancestral birds, such as enantiornithines and *Confuciusornis*, have a precocial development style[5,6,98], although there is an evolutionary transition of a reduced precocity in ornithurine birds[99] and precocity is generally considered to be an anti-predation strategy for facing historically strong predation pressure[100,101]. Moreover, one recent study shows that the CALB was probably cathemeral (i.e., active in both day and night), and that it may have evolved an enhanced visual capability to detect motion[30]. Cathemerality is considered to be linked to high predation risk[102,103] and the promoted motion detection ability of the CALB may mainly help to detect approaching predators[104] given its herbivory. Therefore, the dietary shift may have made ancestral birds become the prey of high-level consumers, possibly leading to their high predation risk.

Knowing the possible predators of ancestral birds is important to determine their potential predation risk. According to arboreal theory, birds evolved from a group of arboreal and gliding maniraptorans, and that ancestral birds may be primarily arboreal and capable of gliding flight, although they spent some time on the ground as well[4–6,10,105,106]. Given the possible arboreality and gliding lifestyle of ancestral birds, while there are many potential predators, such as carnivorous theropods, carnivorous mammals (e.g., *Repenomamus*), snakes (e.g., *Sanajeh*), and crocodylomorphs, in the Mesozoic terrestrial ecosystem[74,86], four lines of evidence may suggest that one group of carnivorous theropods—non-avian maniraptorans (e.g., dromaeosaurids)—is likely one of the main predators of ancestral birds, as proposed previously[5,90,107]. Primarily, a wealth of small feathered non-avian maniraptorans, such as *Aurornis*, *Anchiornis*, *Bambiraptor*, *Buitreraptor*, *Changyuraptor*, *Eosinopteryx*, *Jinfengopteryx*, *Microraptor*, *Rahonavis*, and *Xiaotingia*, are found to have hallmark anatomical characteristics indicative of their capability of gliding flight or even some forms of powered flight[2,4–6,85,88,108,109], and many of these volant non-avian maniraptorans, such as *Microraptor*, *Anchiornis*, and *Changyuraptor*, show predatory features[2,3,8,15,82,86–91,107], representing one of the potential aerial predators of ancestral birds. The predation pressure from these aerial predators may be more significant than those ground predators given the arboreality and gliding lifestyle of ancestral birds. On the other hand, both ancestral birds and gliding non-avian maniraptorans have a relatively small body size among the dinosaurs known[86,95,96,110], suggesting that ancestral birds may

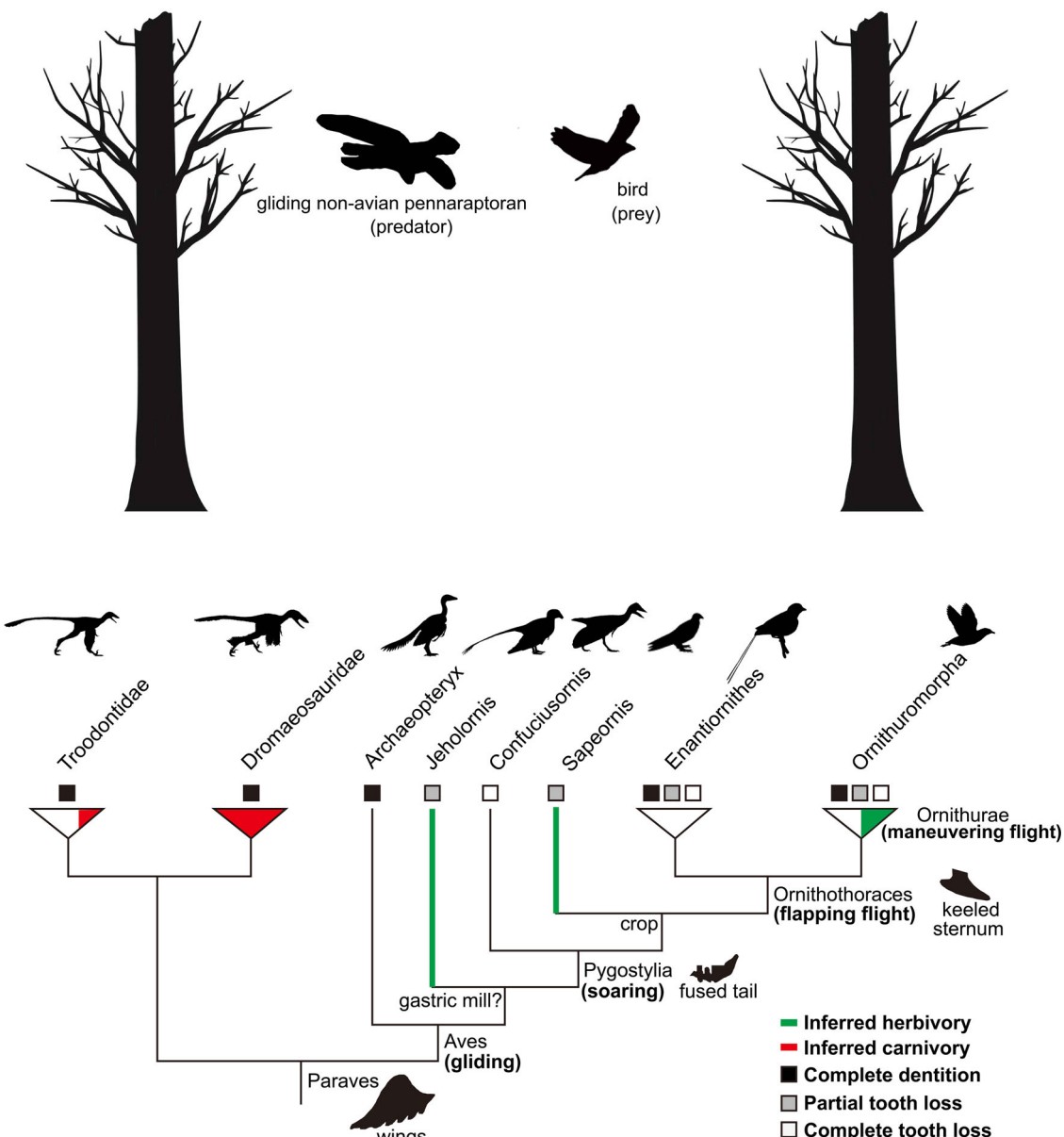

**Fig. 3 Schematic representation of the predation hypothesis underlying the origin of birds proposed in this study.** The predation of gliding predatory non-avian maniraptorans (pennaraptorans) on ancestral birds in the context of the arboreal theory is shown (please see text for details). Paraves phylogeny with digestive system characteristics (gastric mill, crop, and tooth) and taxonomic definition (e.g., Aves) are based on one previous study[3]. The dietary information follows published studies[3,15,82]. The flight-related anatomical features (wings, fused tail, and keeled sternum) along phylogeny follow one published study[146]. The progressive enhancement of flight performance from gliding to soaring, flapping flight, and maneuvering flight within Aves is based on published literature[5]. Species silhouettes corresponding to each of phylogenetic taxa used are from phylopic.org and are designed by (from left to right) the following: Troodontidae (Scott Hartman), Dromaeosauridae (Scott Hartman, modified by T. Michael Keesey), *Archaeopteryx* (Dann Pigdon), *Jeholornis* (Matt Martyniuk), *Confuciusornis* (Scott Hartman), *Sapeornis* (Matt Martyniuk), Enantiornithes (Matt Martyniuk), and Ornithuromorpha (Juan Carlos Jerí).

be a suitable prey for them. This is because there is a general positive correlation of body size between predators and their target prey, and small predators tend to prey on small prey[111–114]. Moreover, previous studies show that, among theropods, non-avian maniraptorans show a relatively high metabolic level (e.g., endothermy) comparable to birds[115,116], suggesting that they possibly had a relatively high activity level. The high activity level of non-avian maniraptorans supports the feasibility of their predation on ancestral birds. Finally, and more importantly, there is already direct fossil evidence indicative of the predation of ancestral arboreal bird (adult enantiornithine bird) by arboreal and

gliding predatory non-avian maniraptorans, such as *Microraptor*, which is known from hundreds of specimens, despite the extreme scarcity of preserved fossils[90]. In addition, the possible predation of ancestral birds by another predatory non-avian maniraptoran, *Sinornithosaurus*, which might be capable of gliding flight[117], was proposed previously[107]. These lines of ecological and fossil evidence suggest that the predation pressure of ancestral birds during their early evolution may, at least partly, mainly have come from those arboreal and gliding non-avian maniraptorans.

The predation from gliding non-avian maniraptorans as described may be one important selection pressure of ancestral

birds, which may have then led to their evolution of anti-predator traits. Among many possible anti-predator traits of ancestral birds, powered flight (e.g., flapping flight) has long been considered to be, at least partly, helpful to escape from predators[5]. Regarding the powered flight of birds, different theories have been proposed to account for its evolution[4,5]. Further, arboreal theory invokes a natural transition of powered flight via gliding flight[4,5,10,105,106], but a basic question remains: what was the selection pressure for the natural transition[118]? Although gliding flight is common among both living and extinct animals, powered flight is rare and is only known in insects, pterosaurs, birds, and bats[119], suggesting that powered flight may less likely occur without certain selection pressures. This is particularly true for birds, as their powered flight demands high energy and substantial evolutionary alternation (e.g., keeled sternum and flight muscles) compared to gliding flight, a simple and cheap way of flying[5]. Early birds, such as *Archaeopteryx* and *Jeholornis*, are believed to be primarily arboreal and be capable of gliding flight, which are believed to be descended from maniraptorans that had already evolved gliding flight[4,5,106]. Indeed, many maniraptorans possess asymmetric flight feathers to generate lift and, in particular, the discovery of many bird-like paravians, such as *Microraptor*, *Anchiornis*, *Xiaotingia*, and *Aurorornis*, is the most unusual in developing four wings, suggesting their possible high performance of gliding flight[4,5,109]. However, given the diet divergence between non-avian maniraptorans and ancestral birds, and particularly that many of gliding non-avian maniraptorans (e.g., *Microraptor* and *Sinornithosaurus*) were potential predators of early birds[5,90,107], it is plausible that early birds may have then evolved powered flight (e.g., flapping flight) based on their gliding flight to escape from gliding predatory non-avian maniraptorans. The predation pressure from gliding predatory non-avian maniraptorans may have worked as a driver to stimulate the evolution of powered flight of their arboreal prey. Moreover, the flapping flight of birds may be critical to flee from those gliding predators. Fossil evidence shows that ever since the evolutionary divergence of early birds from their maniraptoran relatives, the evolution of birds has shown a major trend in the improvement of flight, such as from gliding to flapping and maneuvering flight with the acquisition of flight-related characteristics such as a shortening of the tail and a keeled sternum[2,4,5,85] (Fig. 3). The continuous evolutionary enhancement of the flight of ancestral birds may essentially help for an increase of speed and maneuverability of locomotion, both of which are considered to be crucial for escape success[120]. This may be the case particularly for birds, as they could not become large in body size, a potential anti-predator strategy observed in many animals[121], to evade predators due to their miniaturization constraints in favor of flight[95,96]. Indeed, for many birds, flying is an important means used to escape from predators[5,122], suggesting predation is an important selection pressure for powered flight[5,119]. This is consistent with the observation that birds frequently become flightless in predator-free islands[123]. Thus, the predation pressure from gliding predatory non-avian maniraptorans may be an important candidate contributing to the evolutionary shift from gliding flight to powered flight at the theropod-to-bird transition, although it remains unknown as to whether there were gliding predators other than non-avian maniraptorans contributing to the evolution of powerful flight of birds as well.

Besides the evolutionary specialization of locomotion (e.g., flapping flight), birds have a specialized digestive system. Living birds are toothless and they swallow their food whole, which is temporally stored in their crop and then grinded up by their muscular gizzard. Fossil evidence shows that the specialization of the digestive system occurs in multiple lineages of ancestral

birds[3,6,8,11,92,124–126] (Fig. 3). A recent genomic study shows that modern birds lost their teeth since their common ancestor about 116 million years ago[127]. Regarding the evolutionary specialization of the digestive system of birds, its adaptive significance is, however, less clear. Previous studies indicate that the loss of teeth in birds seems to be linked to an herbivorous diet[5,11,15,81], which is consistent with the herbivory of the CALB found in this study, but the underlying mechanism remains unknown. Optimal foraging theory states that predation has a profound influence on the foraging strategies of animals and animals must trade off two conflicting demands of maximizing foraging efficiency and minimizing predation risk[128–130]. In light of this optimal foraging theory, for the evolutionary specialization of the digestive system of birds, we propose here that herbivores (e.g., ancestral birds) are low-level consumers and, consequently, at relatively high risk to predators. Under high predation risk, the time needed to acquire and process food using the teeth may be limited, but the evolutionary specialization of the digestive system of birds may allow them to gather more food as fast as possible (maximizing foraging efficiency), as food can be stored in their crop without expending too much time processing it using their teeth, and then they can seek a safe place to process their food via their gizzard (minimizing predation risk). Consequently, the reduced reliance on teeth for the processing of food as a result of predator avoidance may have then led to the selection relaxation of the teeth, leading to their subsequent reduction or loss thereof. This may be particularly true for early birds that would necessarily demand frequent foraging[3] and much time for the oral processing of their food (e.g., hard seeds)[8] if no gizzard were available under relatively high predation risk (including both aerial, arboreal, and ground predators), whereas the evolution of the bird-like digestive system may help to maximize foraging efficiency and minimize their exposure to predators. This is consistent with previous studies showing that ancestral birds seem not to have used their teeth to process food; rather, their teeth, if any, were mainly used for the acquisition of food[10,11,79,97,125].

Regarding the reduction or loss of the teeth of birds, it is traditionally attributed to lightening the body for flight[11,126]. This, however, cannot explain the occurrence of numerous toothed Mesozoic birds (e.g., Enantiornithes and *Ichthyornis*)[3,126,131] and hence the teeth were probably not a limiting factor for flight[2,6,97,132]. Alternatively, teeth reduction or loss is considered to be partly due to the functional replacement by the muscular gizzard[3,125,133]. However, this raises a new question: given that teeth and muscular gizzard have a similar function, why the teeth got lost rather than muscular gizzard? One possibility is that it must expend considerable time processing food using teeth without a gizzard during foraging, which may then largely increase their predation risk. In line with this reasoning, the crop is also suggested to help to gather more food quickly, to avoid competitors and/or predators[11,18,125], although an alternative explanation exists[3,97]. Given the possible importance of predation, we argue that the evolution of digestive system characteristics of birds, including teeth reduction or loss, crop, and gizzard, are not independent; rather, their evolution is probably mutually dependent. The integrative and/or collective evolution of these characteristics may be a result of both maximizing foraging efficiency and minimizing predation risk. Predation pressure is also believed to be a potential selection pressure for the evolutionary specialization of the digestive system (e.g., four-chambered stomach) of ruminants[18,134]. Besides birds, teeth reduction or loss is frequently observed in many other tetrapod lineages as well (e.g., toads and turtles)[81,133,135] and future studies would be beneficial to determine whether their teeth reduction or loss was due to historically high predation risks as well.

## Conclusion

Our molecular phyloecological study shows that ancestral birds (e.g., CALB) underwent a dietary shift to be low-level consumers (e.g., fruit, seed, and/or nut eaters), which may have then made them become the prey of potential predators such as gliding non-avian maniraptorans (e.g., dromaeosaurids and troodontids), from which ancestral birds descended. Under this predation pressure, the ancestral birds with inherited gliding flight from their immediate gliding maniraptoran predecessors may have then evolved not only powered flight (e.g., flapping flight) as an anti-predator strategy against gliding predatory non-avian maniraptorans but also a specialized digestive system as an evolutionary tradeoff of maximizing foraging efficiency and minimizing their exposure to predators (including both gliding and non-gliding predators). Our results suggest that dietary shift-associated predation pressure may have facilitated the evolutionary origin of birds.

## Methods

**Taxa used**. We mainly included 95 species in this study. Of the 95 species, 73 species are birds, belonging to 36 orders, representing the majority of living bird orders (36/39)[136] and 22 species are relatives of birds, including 5 crocodilians, 6 turtles, and 11 squamates (Fig. 1). For the 73 bird species included, the majority come from Neognathae, with relatively little species of Palaeognathae. For the Palaeognathae species included, the GenBank sequences of many of our focal genes were missing upon our initial sequence analyses, especially for the ostrich (*Struthio camelus*) and the emu (*Dromaius novaehollandiae*), and thus we selected the two species for transcriptome sequencing.

**Sampling, RNA isolation, and cDNA library construction**. One ostrich (3 months old) and one emu (6 months old) were used for sampling. The two individuals were the same two individuals used in one of our previous studies[30]. Further, the methods of RNA isolation and cDNA library construction were almost identical to those of that study[30]. Briefly, the two active individuals of an artificial breeding company (Quanxin, Daqing) were transported to the laboratory with vegetables and water provided. The two individuals were killed after 24 h and an approximately equal amount of tissue from the liver, pancreas, stomach (proventriculus), and small intestine (duodenum) were obtained and mixed. The mixed tissues were preserved in RNA-locker (Sangon Biotech, Shanghai), flash frozen in liquid nitrogen, and then transferred to a −80 °C refrigerator until further processing. The experimental procedures were carried out following an animal ethics approval granted by Northeast Normal University. All experimental procedures in this study were approved by the National Animal Research Authority of Northeast Normal University, China (approval number: NENU-20080416) and the Forestry Bureau of Jilin Province of China (approval number: [2006]178).

We isolated the total RNA of the two samples using TRIzol Reagent (Invitrogen Life Technologies), following the manufacturer's protocol and instructions. We monitored the RNA degradation and contamination on 1% agarose gels. We checked the RNA purity using the NanoPhotometer® spectrophotometer (IMPLEN, CA, USA). We measured RNA concentration using the Qubit® RNA Assay Kit in a Qubit®2.0 Flurometer (Life Technologies, CA, USA). We assessed the RNA integrity using the RNA Nano 6000 Assay Kit of the Agilent Bioanalyzer 2100 system (Agilent Technologies, CA, USA). We constructed the cDNA library using NEBNext®Ultra™ RNA Library Prep Kit for Illumina® (NEB, USA) according to the manufacturer's protocol. Accordingly, mRNA was purified using poly-T oligo-attached magnetic beads. The enriched mRNA was fragmented into small pieces and was then used for the syntheses of the cDNA strands. The cDNA was purified and size-selected using the AMPure XP system (Beckman Coulter, Beverly, MA, USA). Then, PCR analysis was performed and the PCR products were purified (AMPure XP system). Library quality was assessed on the Agilent Bioanalyzer 2100 system. Paired-ending sequencing was performed using Illumina HiSeq X-ten (Biomarker Technology Co., Beijing).

**Data filtering and de novo assembly**. We generated 10.47 and 10.45 G bases for the ostrich and emu, respectively. We filtered the raw data by removing reads containing adapters, reads containing ploy-N, and low-quality reads. Clean reads were assembled using the de novo assembly program Trinity (v2.5.1)[137] with default parameters. Unigenes were generated and unigenes longer than 200 bp were retained for subsequent analyses.

**Genes used and sequence alignment**. The genes annotated in three KEGG digestive system pathways, including CDA (map04973), PDA (map04974), and FDA (map04975), were used in this study (Fig. 2 and Supplementary Data 3). For these digestive system-related genes, we abstracted their sequences from the studied ostrich and emu. For this, we downloaded the coding sequences of our target genes

of *Gallus gallus* from GenBank and used them as query sequences to blast against the unigene pools of the two species using Blastn software. We subsequently annotated these unigene sequences returned by blasting against the NCBI nr/nt database using the online Blastn and we kept only the unigene sequences with the same gene annotation as that of the query sequences for subsequent analyses. Besides the two species, we also downloaded our target gene sequences from all birds and reptiles with gene sequences available in GenBank (Supplementary Data 1). For five genes (e.g., *CPA2*, *G6PC*, *PLA2G2E*, *SLC2A5*, and *SLC36A1*), their sequences of the reptile relatives of birds were unavailable and, thus, their sequences from mammals and amphibians were used. For our focal genes, those genes (e.g., amylase genes) with sequences unavailable or available for only few bird species were excluded from our analyses and, eventually, 83 genes were retained for subsequent analyses. We aligned gene sequences using webPRANK (http://www.ebi.ac.uk/goldman-srv/webprank/) and individual sequences with long indels and/or lengths that were too short were removed or replaced by other relevant transcript variants. After this pruning, the translated protein sequences of these genes were blasted against the non-redundant protein sequence database to confirm the correctness of the sequence cutting.

**Positive selection analyses**. We performed positive selection analyses of genes using branch and branch-site models implemented in the Codeml program of PAML[33]. For this, an unrooted species tree (Fig. 1) was constructed following published studies[30,138–143], with the phylogenetic relationships among bird orders following one genome-level study[144]. We estimated the ratio of non-synonymous to synonymous substitutions per site (dN/dS or $\omega$) and employed likelihood ratio tests (LRTs) to determine statistical significance. Positive selection is determined by the value of $\omega > 1$ with a statistical significance. Bonferroni multiple testing correction was used to adjust *p*-values.

**Branch model**. We performed positive selection analyses of genes along our focal branches using a two-rate branch model. Upon analysis, we labeled our focal branches as foreground branches and the rest were used as background branches. For this model, $\omega$ is assumed to be different between foreground branches and background branches, and its goodness-of-fit was analyzed using the LRT by comparing it with the one-rate branch model that assumes a single $\omega$-value across all branches. If a statistically significant value of $\omega > 1$ was detected in a foreground branch, the two-ratio branch model was further compared with the two-ratio branch model with a constraint of $\omega = 1$ of the foreground branch to further determine whether the value of $\omega > 1$ of the foreground branch was supported with statistical significance.

**Branch-site model**. We also used a branch-site model (Test 2) to detect positive selection genes for our focal branches. The branch-site model assumes four classes of sites, with site class 0 and site class 1, respectively, representing evolutionarily conserved ($0 < \omega_0 < 1$) and neutral codons ($\omega_1 = 1$) across branches, and site classes 2a and 2b representing evolutionarily conserved ($0 < \omega_0 < 1$) and neutral ($\omega_1 = 1$) codons for background branches, yet allowed to be under positive selection ($\omega_2 > 1$) for the foreground branches. The goodness-of-fit of this model was analyzed by using the LRT, by comparing a modified model A with a null model with $\omega = 1$ constrained. Positively selected sites were analyzed by an empirical Bayes method.

**Selection intensity analyses**. The selection intensity changes of genes were evaluated using RELAX[62], which is available from the Datamonkey webserver (http://test.datamonkey.org/relax). For the selection intensity analyses, a parameter *k*-value and its statistical significance were estimated given a priori partitioning of test branches and reference branches in a codon-based phylogenetic framework. Intensified selection is indicated by $k > 1$ and is expected to exhibit $\omega$ categories away from neutrality ($\omega = 1$), whereas a relaxed selection is indicated by $k < 1$ and is expected to exhibit $\omega$ categories converging to neutrality ($\omega = 1$). The statistical significance of the *k*-value was evaluated by comparing an alternative model to a null model using LRT, with the former assuming different $\omega$ distributions of the test and reference branches, and the latter assuming $k = 1$ and the same $\omega$ distribution of both test and reference branches.

**Reporting summary**. Further information on research design is available in the Nature Research Reporting Summary linked to this article.

## Data availability

The transcriptome sequencing data were deposited into the National Center for Biotechnology Information Sequence Read Archive database under accession numbers (SRR12237019-20). All other data needed to evaluate the conclusions in the paper are present in the paper and/or the Supplementary Materials, or are available from the corresponding author on reasonable request.

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

## Acknowledgements

We thank Lin Chen, Yuanqin Zhao, and Li Gu for helping tissue sampling. This research was supported by the National Natural Science Foundation of China (grant number 31770401) and the Fundamental Research Funds for the Central Universities.

## Author contributions

Y.W. designed research, performed analyses, and wrote the paper.

## Competing interests

The author declares no competing interests.
