## [Peer Review File · Communications Biology]

Reviewers' comments:

Reviewer #1 (Remarks to the Author):

In this manuscript, the author largely used available genome sequences and identified 83 digestive system-related genes in birds and other reptiles. Inferred from the results of selective pressure analyses, the author attempted to revisit two big questions: trophic shifts in archosaurs, and the origin of birds. The 83 genes came from the three pathways related to carbohydrate digestion and absorption (CDA), protein digestion and absorption (PDA), and fat digestion and absorption (FDA), the author thus made many suggestions of trophic shifts in archosaurs and the origin of birds based on his/her selection pressure analyses. The two big questions are topics of very broad interest, but I think the author extended his/her data too far to explain trophic shifts and the origin of birds.

1) My major concern is that the genetic data are too limited. The three pathways can not represent all genes involved in trophic shifts and the origin of birds, and thus the conclusions of this paper are unwarranted. The author is using very small data sets to address very big questions.

2) The writing is unclear. I can not follow how many species s/he used. I don't know how s/he adjusted the p-values of likelihood ratio tests because the authors tested same data sets many times. I thus can not assess how significant his/her results.

3) Even positive selection and selection intensity are statistically significant, one can not extend too far when interpreting the data, because many evolutionary events have shaped these selection patterns, how can you rule out other possibilities that have led to the current regimes of selection.

4) Lines 105-111, when you compare data among three groups, how come you said two groups are the most predominant? What is the cutoff of the most predominant positive selection signals? Did you adjust the p-values generated from multiple testing?

5) Line 152, I don't think you demonstrate anything here.

6) I did not finish reading the Discussion, I think this part contains too many speculations without solid evidence.

7) Line 466, sequence abstraction is not a commonly used term, please consider alternative terms.

8) Line 489, one should not use the Tree of Life Web Project as the species tree, instead we should follow the papers cited in the Tree of Life Web Project, because this web project often generates confusing tree topologies, due to inconsistency in current literature.

9) Line 493, one should write how you corrected the p-values generated from multiple testing.

Reviewer #2 (Remarks to the Author):

The MS "Trophic shift and the origin of birds" is well organized and discussed, and the figures are open-and-shut. The authors employed the molecular phyloecological approach using digestive system-related genes as molecular markers to infer the diets of the ancestral archosaur and of the common ancestor of living birds (CALB). The results suggest that the powered flight and specialized digestive system of birds may have evolved as a result of their trophic shift-associated predation pressure. In my opinion, this paper is within the publication extent of Communications Biology. The following are some minor revisions.

(1) In the introduction section, add some description of digestive enzyme genes.

(2) Add some important tables in the maintext.

Reviewer #3 (Remarks to the Author):

Dear Dr. Wu,

Thank you for the chance to review your work. Overall, I think the premise and results of your paper are very good, but the discussion needs significant work before I can recommend publication.

Firstly, I see a major alternate explanation for your results that needs addressing: flight also has a direct effect on digestive enzymes and efficiency, as seen in birds and bats (e.g. Caviedes-Vidal et al. 2007 and Frei et al. 2014). In particular, flighted organisms tend to evolve shorter gut retention times and higher nutrient assimilation rates to minimize the time that digesting food weighs them down. The digestive physiology results could also be interpreted as an adaptation for shorter gut retention time; fats and complex carbohydrates tend to digest slower than proteins, so early birds may have favored quicker processing of the slower-digesting molecules for faster ejection. This alternative explanation needs to be mentioned and discussed.

Secondly, a lot of your discussion isn't really related to your results. The subsections "Trophic shift of ancestral birds as low-level consumers", "Main predators of ancestral birds—gliding maniraptorans", and "Flapping flight as a possible anti-predator strategy against gliding maniraptorans" are more like summaries of the literature than elaborations on the significance of your findings. I suggest either finding ways to refer back to the results section and make the links to your new data clearer or simply condensing the sections into the introduction or the remaining discussion subsections.

Thirdly, you have some inaccuracies relating to the citations you have used:

Page 8 refers to bite force without any citations. Ma et al.'s volume chapter addresses what you are looking at it, but finds the trend you claim is indicative of dinosaur-bird transition is at the paravian node. Please cite Ma et al. papers:

Ma, W., Pittman, M., Lautenschlager, S., Meade, L.E., and Xu, X. (2020). Functional morphology of the oviraptorosaurian and scansoriopterygid skull. In Pennaraptoran theropod dinosaurs: past progress and new frontiers, M. Pittman and X. Xu, eds. (New York: Bulletin of the American Museum of Natural History), pp. 229-249.

Ma, W.S., Wang, J.Y., Pittman, M., Tan, Q.W., Tan, L., Guo, B., and Xu, X. (2017). Functional morphology of a giant toothless mandible from a bird-like dinosaur: Gigantoraptor and the evolution of the oviraptorosaurian jaw. *Scientific Reports* 7, 16247.

Page 10 asserts all early birds as precocial, but according to Mayr 2017 the closer to the CALB the more altricial they get. Please cite Mayr 2017.

Other points:

Page 13 uses circular logic: fossilized whole foods mean birds didn't use their teeth to process food, but using teeth to process food would make the food less likely to fossilize. There is evidence of eating whole, but not evidence for absence of dental processing. Please edit.

Page 14 gives early birds as an example for edentulism indicating herbivory, but you do not state if they are basing that off of their paper's work (as the literature is divided, some not proposing herbivory as the ancestral state).

Table 2 differs from Wang et al 2020 volume chapter on what clades have what levels of tooth loss. This needs to be corrected and updated.

Important general points:

The paper is under-referenced. I have added many key references. There is also a tendency to rely on overly general references. Please take the opportunity to improve referencing in the paper.

Please use the correct taxonomic language so that the reader knows exactly what groups are being referred to. For example, avialans, non-avian avialans, avians etc. Please consult and cite: Pittman, M., O'Connor, J., Field, D.J., Turner, A.H., Ma, W., Makovicky, P., and Xu, X. (2020). Pennaraptoran systematics. In Pennaraptoran theropod dinosaurs: past progress and new frontiers, M. Pittman and X. Xu, eds. (New York: Bulletin of the American Museum of Natural History), pp. 7-36.

I hope this feedback has been helpful and I look forward to seeing a revised version of your work.

Best regards,

Michael Pittman
The University of Hong Kong

Referee expertise:

Referee #1: Molecular evolution, comparative genomics, phylogenetics, digestive enzymes

Referee #2: Evolution of digestive enzyme genes

Referee #3: Avian and flight origins

Reviewers' comments:

Reviewer #1 (Remarks to the Author):

In this manuscript, the author largely used available genome sequences and identified 83 digestive system-related genes in birds and other reptiles. Inferred from the results of selective pressure analyses, the author attempted to revisit two big questions: trophic shifts in archosaurs, and the origin of birds. The 83 genes came from the three pathways related to carbohydrate digestion and absorption (CDA), protein digestion and absorption (PDA), and fat digestion and absorption (FDA), the author thus made many suggestions of trophic shifts in archosaurs and the origin of birds based on his/her selection pressure analyses. The two big questions are topics of very broad interest, but I think the author extended his/her data too far to explain trophic shifts and the origin of birds.

1) My major concern is that the genetic data are too limited. The three pathways can not represent all genes involved in trophic shifts and the origin of birds, and thus the conclusions of this paper are unwarranted. The author is using very small data sets to address very big questions.

Reply: We understand your concern about this, and we respond to your concern below for your kind consideration. We in this study used the genes involved in three KEGG pathways related to carbohydrate, protein, and fat digestion and absorption as the molecular markers indicative of diets to infer the diet of our focal taxa. Digestive physiology studies (e.g., Karasov et al. 2011; Karasov and Douglas 2013) and recent studies (e.g., Chen and Zhao 2019; Wang et al. 2020b; Wu et al. 2020) on the adaptive evolution of digestive system-related genes have shown that digestive system-related genes are capable of reflecting the dietary variations of animals. And accordingly, our molecular results provide relatively clear evidence about the diets of ancestral archosaurs and ancestral birds, which are highly consistent with fossil evidence. The diet reconstruction provides important insights into understanding trophic shifts and the origin of birds. Regarding the trophic shifts and the origin of birds, as you said, it is related to many traits with relevant genes (e.g., beak, feather, wing and endothermy) besides the digestive system-related genes examined in this study. Examining the adaptive evolution of these traits-related genes may help to determine their possible positive selection in ancestral archosaurs and ancestral birds, but it cannot tell us about their trait states as the molecular markers indicative of these trait states remain to be explored. So far, the molecular markers indicative of diel activity patterns and diets have only been known, while the possible molecular markers indicative of other relevant traits such as beak, feather, wing and endothermy related to the trophic shifts and the origin of birds remain largely unknown. Once these molecular markers are developed in future, it could be helpful to determine the trait states of our focal taxa in the context of molecular method that we used. We could do this as relevant molecular markers are developed in future (This is our goal!). Though all the genes related to the trophic shifts and the origin of birds cannot be studied in our study, we cite fossil and ecological evidence besides our molecular findings to support the possibility of trophic shift-associated predation pressure as a possible driver underlying the evolution of powerful flight and digestive system specialization of birds. To be honest, our molecular finding of the herbivory of ancestral birds itself tells us nothing about the evolution of bird characteristics (flapping flight, toothless, crop, and gizzard), while the finding of the herbivory, combined with ecological and fossil evidence, do support the predation hypothesis that we

proposed to explain the origin of these bird characteristics.

2) The writing is unclear. I can not follow how many species s/he used. I don't know how s/he adjusted the p-values of likelihood ratio tests because the authors tested same data sets many times. I thus can not assess how significant his/her results.

Reply: Done! We in our revised manuscript add the species number used for the analyses of each positively selected gene found, and the p values under Bonferroni multiple testing correction is also added. Please see Table 1 for details. The species number of each gene is under change due to the availability of their gene sequences (please see Table S1 for the number of species sequences used for all genes examined). The positive selection signals remain largely unchanged after Bonferroni multiple testing correction.

3) Even positive selection and selection intensity are statistically significant, one can not extend too far when interpreting the data, because many evolutionary events have shaped these selection patterns, how can you rule out other possibilities that have led to the current regimes of selection.

Reply: The positive selections of digestive system genes are normally explained to be linked to the diets in previous studies. Though about this, we agree with you that there may be different possibilities to explain the selection patterns found in our study. Considering that previous studies show that, in addition to diets, flight and microbial fermentation may be two other important factors to affect the evolution of digestion system genes (Caviedes-Vidal et al. 2007; Frei et al. 2015; Price et al. 2015), we add one additional paragraph to discuss these possibilities. Though about this, we find that diet is remained as one main explanation for our molecular results, consisting with the findings in digestive physiology (Karasov et al. 2011; Karasov and Douglas 2013). Please see lines 214 to 236 for details.

4) Lines 105-111, when you compare data among three groups, how come you said two groups are the most predominant? What is the cutoff of the most predominant positive selection signals? Did you adjust the p-values generated from multiple testing?

Reply: "The most predominant" is an incorrect word here as you indicate. We change it to "relatively strong" in our revised manuscript. We also check throughout our manuscript and get it revised. Please see line 110 and line 114. We also add the corrected p values of positively selected genes through Bonferroni multiple testing correction. Please see Table 1, and lines 186 to 187 for details.

5) Line 152, I don't think you demonstrate anything here.

Reply: The sentence is edited to read like "Our positive selection analyses described above showed that the CALB exhibited a predominant Darwinian selection of the genes related to carbohydrate and fat digestion and absorption, whereas the ancestral archosaur exhibited a marked Darwinian selection of the genes related to protein and fat digestion and absorption, indicating substantial selection differences between them." Please see lines 156 to 159 for detail.

6) I did not finish reading the Discussion, I think this part contains too many speculations without solid evidence.

Reply: We in our Discussion Section discussed important implication of the herbivory in the origin of flapping flight and digestive system of birds based on fossil evidence. And the combined data of our molecular and fossil findings suggest the potential role of predation in the origin of birds. For this, we cite relevant ecological and paleobiological papers to support this possibility. We agree with you that there were some speculations without solid evidence in our original manuscript version. To strengthen the findings of our study, we check throughout our manuscript several times to remove

those sentences without solid evidence and only remain those with relevant supports of evidence (literature). The sentences listed below are removed in our revised manuscript.

“Possibly, there was some form of coevolution between ancestral birds and those gliding predatory maniraptorans.”

“It appears that all the tetrapod lineages with teeth reduction or loss tend to be low-level consumers (e.g., herbivores and insectivores), at least during their early evolution, as exemplified by ancestral birds, and hence may have been faced with relatively high predation pressures. This may imply that high predation pressure might be a common force in facilitating the reduction or loss of teeth. Of course, there are many low-level consumers (e.g., herbivorous dinosaurs) that still retained their teeth. This might be partly attributed to their lineages’ specific low-level predation pressures due to their, for instance, efficient anti-predator strategies (e.g., large body size). In addition to predation pressure, the possible effects of other ecological factors on the evolution of teeth may exist, such as, food types that may determine the reliance of food acquisition and processing on teeth. Most likely, the evolutionary dynamic of teeth may partly reflect a tradeoff between the selections to maintain or strengthen teeth (e.g., acquiring and processing food) and the selections to reduce teeth as a by-product of accelerating feeding to avoid predators.”

7) Line 466, sequence abstraction is not a commonly used term, please consider alternative terms.

Reply: Done! The subtitle is edited as “Genes used and sequence alignment”. Please see line 496.

8) Line 489, one should not use the Tree of Life Web Project as the species tree, instead we should follow the papers cited in the Tree of Life Web Project, because this web project often generates confusing tree topologies, due to inconsistency in current literature.

Reply: Done! We follow the original paper (Jønsson and Fjeldså 2006) cited in the Tree of Life Web Project, and the citation of Tree of Life Web Project is removed from our revised version. Please see lines 517 to 520.

9) Line 493, one should write how you corrected the p-values generated from multiple testing.

Reply: Done! We add one sentence here to read “Bonferroni multiple testing correction was used to adjust p values.” Please see line 523.

Reviewer #2 (Remarks to the Author):

The MS “Trophic shift and the origin of birds” is well organized and discussed, and the figures are open-and-shut. The authors employed the molecular phyloecological approach using digestive system-related genes as molecular markers to infer the diets of the ancestral archosaur and of the common ancestor of living birds (CALB). The results suggest that the powered flight and specialized digestive system of birds may have evolved as a result of their trophic shift-associated predation pressure. In my opinion, this paper is within the publication extent of *Communications Biology*. The following are some minor revisions.

(1) In the introduction section, add some description of digestive enzyme genes.

Reply: Done! Thank you for the suggestion. We add some description of the genes in our revised manuscript. Considering that a lot of genes were used, and we add a summary of the function description of these genes at the beginning of our Result section, reading like “The 83 genes came from three digestive system-related KEGG pathways including carbohydrate digestion and

absorption (CDA), protein digestion and absorption (PDA), and fat digestion and absorption (FDA) (Fig. 2). The functions of these digestive system-related genes are relatively well-studied and are known to play important roles in the digestion and absorption of carbohydrates, proteins, and fats." Please see lines 98 to 102.

(2) Add some important tables in the maintext.

Reply: Done! Thank you for the suggestion. Two tables (Table S2 and Table S3) about the positive selection genes of our original manuscript version are moved to our maintext as Table 1 in the revised version. Please see lines 966 to 970.

Reviewer #3 (Remarks to the Author):

Dear Dr. Wu,

Thank you for the chance to review your work. Overall, I think the premise and results of your paper are very good, but the discussion needs significant work before I can recommend publication.

Firstly, I see a major alternate explanation for your results that needs addressing: flight also has a direct effect on digestive enzymes and efficiency, as seen in birds and bats (e.g. Caviedes-Vidal et al. 2007 and Frei et al. 2014). In particular, flighted organisms tend to evolve shorter gut retention times and higher nutrient assimilation rates to minimize the time that digesting food weighs them down. The digestive physiology results could also be interpreted as an adaptation for shorter gut retention time; fats and complex carbohydrates tend to digest slower than proteins, so early birds may have favored quicker processing of the slower-digesting molecules for faster ejection. This alternative explanation needs to be mentioned and discussed.

Reply: Done! Thank you very much for the suggestion. We do think that it is important to discuss the possible effects of flight on our result explanation, which we neglected in our original manuscript. For this, we add one additional paragraph with the citation of previous studies (Caviedes-Vidal et al. 2007; Frei et al. 2015; Price et al. 2015), including the two papers that you recommended. Overall, the possible effects of flight on our results can be largely excluded mainly because that flight has been found to almost only affect the nutrient utilization related to the paracellular absorption pathway rather than the transcellular absorption pathway, which includes the carbohydrate, protein, and fat digestion and absorption KEGG pathways that we examined in our study. In addition to flight, we also discuss the possible effects of microbial fermentation on our results explanation. Please see lines 214 to 236 for details.

Secondly, a lot of your discussion isn't really related to your results. The subsections "Trophic shift of ancestral birds as low-level consumers", "Main predators of ancestral birds—gliding maniraptorans", and "Flapping flight as a possible anti-predator strategy against gliding maniraptorans" are more like summaries of the literature than elaborations on the significance of your findings. I suggest either finding ways to refer back to the results section and make the links to your new data clearer or simply condensing the sections into the introduction or the remaining discussion subsections.

Reply: Done! We modify or edit some sentences at the beginning of each of the three sections that you mentioned to either refer back to our results or make logic links between them. Please see lines 269 to 270, 309 to 312, 349 to 353 for details.

Thirdly, you have some inaccuracies relating to the citations you have used:

Page 8 refers to bite force without any citations. Ma et al.'s volume chapter addresses what you are

looking at it, but finds the trend you claim is indicative of dinosaur-bird transition is at the paravian node. Please cite Ma et al. papers:

Ma, W., Pittman, M., Lautenschlager, S., Meade, L.E., and Xu, X. (2020). Functional morphology of the oviraptorosaurian and scansoriopterygid skull. In Pennaraptoran theropod dinosaurs: past progress and new frontiers, M. Pittman and X. Xu, eds. (New York: Bulletin of the American Museum of Natural History), pp. 229-249.

Ma, W.S., Wang, J.Y., Pittman, M., Tan, Q.W., Tan, L., Guo, B., and Xu, X. (2017). Functional morphology of a giant toothless mandible from a bird-like dinosaur: Gigantoraptor and the evolution of the oviraptorosaurian jaw. *Scientific Reports* 7, 16247.

Reply: Done! Thank you very much for the recommendation of the two papers. Regarding the two papers, we found Ma et al. (2020) is more related to our study as it demonstrates the decrease of biting force from dromaeosaurids to early avialans (including ancestral birds). This is largely consistent with that found by Li et al. (2020). Ultramicrostructural reductions in teeth: implications for dietary transition from non-avian dinosaurs to birds *BMC Evol Biol* 20:46. Actually, we cited Li et al. (2020) in our original manuscript to show the decrease of biting force in birds. For the revised manuscript, we cite both Li et al. (2020) and Ma et al. (2020). Please see line 256.

Page 10 asserts all early birds as precocial, but according to Mayr 2017 the closer to the CALB the more altricial they get. Please cite Mayr 2017.

Reply: Done! The sentence referred is edited and we cite Mayr 2017 (*Evol Ecol* 31:131–141) in our revised version. It reads like “In support of this, fossil evidence shows that ancestral birds, such as enantiornithines and Confuciusornis, have a precocial development style (Zhou and Zhang 2004; Chatterjee 2015; Chiappe and Qingjin 2016) though there is an evolutionary transition of a reduced precocity in ornithurine birds (Mayr 2017b)”. Please see lines 297 to 300 for details.

Other points:

Page 13 uses circular logic: fossilized whole foods mean birds didn't use their teeth to process food, but using teeth to process food would make the food less likely to fossilize. There is evidence of eating whole, but not evidence for absence of dental processing. Please edit.

Reply: Done! We agree with you about this, and we edit the sentence to read like “This is consistent with previous studies showing that ancestral birds seem not to have used their teeth to process food; rather, their teeth, if any, were mainly used for the acquisition of food (Zhou and Zhang 2002; Zheng et al. 2011; Zheng et al. 2014; O'Connor and Zhou 2015; Li et al. 2020)”. Please see line 417 for details.

Page 14 gives early birds as an example for edentulism indicating herbivory, but you do not state if they are basing that off of their paper's work (as the literature is divided, some not proposing herbivory as the ancestral state).

Reply: Done! The section referred is removed due to the lack of solid evidence based on the comments of the first reviewer.

Table 2 differs from Wang et al 2020 volume chapter on what clades have what levels of tooth loss. This needs to be corrected and updated.

Reply: We read throughout the paper (Wang et al 2020), and found it shows detailed information about levels of tooth loss in Maniraptoriformes including birds, especially the tooth loss in the

premaxilla, maxilla and dentary. This is very similar to that of one paper (O'Connor 2019 *Palaeogeography, Palaeoclimatology, Palaeoecology* 513: 178–195) that we originally cited in our paper. However, we find Wang et al 2020 does not show the teeth data of Dromaeosauridae and Troodontidae. Moreover, we realized that tooth reduction patterns are fairly complicated as tooth loss may show various patterns in premaxilla, maxilla and dentary, and it is inconvenient to show these complicated patterns in our figure (Fig.3). And meanwhile, we think it may be meaningless to show the complicated patterns of tooth loss in our study and the simplified pattern of tooth loss that we used is informative. Considering about this, we would keep our Fig.3 unchanged. We cite Wang et al 2020 in our revised version regarding the tooth loss and reduction in birds.

Important general points:

The paper is under-referenced. I have added many key references. There is also a tendency to rely on overly general references. Please take the opportunity to improve referencing in the paper.

Reply: Done! Thank you for your recommendation of relevant references for our manuscript, and accordingly, we add relevant literature including almost all papers that you recommended to us in our revised manuscript. Moreover, regarding the general references (books, e.g., Benton 2015; Mayr 2017; Chatterjee 2015), we cited them many times in our manuscript because these books summarized different aspects of the evolution of birds, which are related to our study. However, we also cited relevant specific literature as needed besides the general references. For the revised version, we checked throughout our literature cited, and add or adjust the citations of some literature to improve referencing of our manuscript. Please see references highlighted in red in our main text for these changes. Thank you for the suggestion.

Please use the correct taxonomic language so that the reader knows exactly what groups are being referred too. For example, avialans, non-avian avialans, avians etc. Please consult and cite: Pittman, M., O'Connor, J., Field, D.J., Turner, A.H., Ma, W., Makovicky, P., and Xu, X. (2020). Pennaraptoran systematics. In *Pennaraptoran theropod dinosaurs: past progress and new frontiers*, M. Pittman and X. Xu, eds. (New York: Bulletin of the American Museum of Natural History), pp. 7-36.

Reply: We read throughout the paper, a comprehensive review about the phylogeny of Pennaraptoran theropod dinosaurs. We find the taxonomic definition (e.g., Aves) in your paper is somewhat different from that of O'Connor 2019 (*Palaeogeography, Palaeoclimatology, Palaeoecology* 513: 178–195), which we cited in our original manuscript. Considering that the Maniraptoran phylogeny with digestive system characteristics (gastric mill, crop, and tooth) and taxonomic definition (e.g., Aves) used in our study (Fig.3) originally follows O'Connor 2019, we revised throughout our manuscript (including Fig.3) mostly using non-avian maniraptoran as needed. This revision may help to know exactly what groups were referred in our manuscript. Thank you for the good suggestion. While we could not cite the paper you recommended, we cite another paper of yours in our revised manuscript (Pittman, M., J. O'Connor, E. Tse, P. Makovicky, D. J. Field, W. Ma, A. H. Turner, M. A. Norell, R. Pei, and X. Xu. 2020. The fossil record of Mesozoic and Paleocene pennaraptorans. Pp. 37-95 in M. Pittman, and X. Xu, eds. *Pennaraptoran Theropod Dinosaurs: Past Progress and New Frontiers*. Scientific Publications of the American Museum of Natural History, New York.). Please see line 290.

I hope this feedback has been helpful and I look forward to seeing a revised version of your work.

Best regards,
Michael Pittman
The University of Hong Kong

Reviewers' comments:

Reviewer #1 (Remarks to the Author):

The author has done a great deal of work to fully address my comments. I think the revised manuscript has been substantially improved. I have no more comments, and I am happy to recommend acceptance without further delay.

Reviewer #2 (Remarks to the Author):

The authors have carefully revised the manuscript according to the majority of the comments previously raised.

Reviewer #3 (Remarks to the Author):

Dear Dr. Wu (Hi Yonghua),

Thank you for making a range of changes to your manuscript, which have improved its quality. However, there are a number of revisions that I feel are necessary for me to support publication.

The use of *Aves* is debated, but most theropod palaeontologists use *Aves* to refer to the crown group. It is not wrong to use it as in O'Connor (2019) but you need to explicitly give the definition you are using in the introduction of your paper to avoid confusing readers that use *Aves* in different ways: I use *Aves* to mean...(sensu O'Connor 2019 which are based on XXXX). It is crucial to refer to the original definition (XXXX) and not just O'Connor (2019) as people won't know the definitions used in O'Connor (2019) from the reference but they will know if you list the original studies. Please see this chapter for a recent example of how to do this: Pittman, M., O'Connor, J., Field, D.J., Turner, A.H., Ma, W., Makovicky, P., and Xu, X. (2020). Pennaraptoran systematics. In Pennaraptoran theropod dinosaurs: past progress and new frontiers, M. Pittman and X. Xu, eds. (New York: Bulletin of the American Museum of Natural History), pp. 7-36. This is important to do --- otherwise different people will think you are talking about different things in your paper.

The section "Main predators of ancestral birds—gliding non-avian maniraptorans" is still disjointed from the rest of the paper. You seem to use the paragraph to establish that non-avian paravians could have fed on avialans, and thus it sets up the following paragraph. If this is your intention, please help the reader by explicitly spelling out that this out is what the paragraph is doing -- now the first two lines talk about avialans being prey and the rest of the paragraph focuses on predatory characters of non-avian avialans so it is confusing to read.

One important reference to add is Miller et al. 2020 which was just published in *Communications Biology* and is directly related to lines 240-241. This reference is also worth adding to the citation in lines 50-52 and adding *Confuciusornis* to the preceding sentence:

Miller, C. V., Pittman, M., Kaye, T. G., Wang, X., Bright, J. A., & Zheng, X. (2020). Disassociated rhamphotheca of fossil bird *Confuciusornis* informs early beak reconstruction, stress regime, and developmental patterns. *Communications Biology*, 3(1), 1-6.

Figure 3 needs more work. Figure 3 includes Maniraptora but this is the wrong ancestor to talk about. It should be replaced with the more inclusive Pennaraptora node as this includes Oviraptorosauria which is the only other group of vane feathered non-paravian theropods. Many oviraptorosaurians are beaked and are thought to be herbivorous so this is highly relevant to your paper and should be marked in this figure (see Pittman & Xu 2020 Pennaraptoran theropod dinosaurs volume). Oviraptorosaurians are important to mark in Figure 3 and to mention -- at least briefly -- in the main text because non-paravian maniraptorans covers far too many groups that you do not mention in the paper and most of them aren't even covered in vane feathers. In

that context, the Ma et al. (2017) is important to cite in your paper as a important paper on oviraptorosaurian diet:

Ma, W.S., Wang, J.Y., Pittman, M., Tan, Q.W., Tan, L., Guo, B., and Xu, X. (2017). Functional morphology of a giant toothless mandible from a bird-like dinosaur: Gigantoraptor and the evolution of the oviraptorosaurian jaw. *Scientific Reports* 7, 16247.)

Whilst you want to keep Figure 3 simple, it is still misleading as it is. Saying a taxon with a fully edentulous jaw as having "partial tooth loss" is not correct. Please correct the figure as the reader must currently read Wang et al. (2020) first to get the correct and full story of tooth which is not appropriate. You will not need to reproduce Wang et al. 2020 in full but the current Figure 3 needs to be improved or readers will think it contradicts recent work by a specialist on theropod tooth loss.

Regarding alternative explanations for the results, the new paragraph is much better. For microbial fermentation, there is one bird to my knowledge (*Opisthocomus hoazin*) which does exhibit microbial fermentation of its food, and Alan Feduccia proposes it may have been present in enantiornithines (Feduccia 1999 page 150). I suggest looking at his evidence and deciding for yourself if this possibility is worth discussing further.

Feduccia, A. (1999). *The origin and evolution of birds*. 2nd edition. Yale University Press.

This Miller and Pittman (2020) preprint (<https://www.essoar.org/doi/10.1002/essoar.10504068.2>) provides a much more comprehensive synthesis of avialan diet than O'Connor (2019), and thus should be cited in the citation lists of proposed bird diets as it is a DOI publication. The preprint is awaiting post-review acceptance in *Biological Reviews* which is expected in January 2020.

I hope you can make these final requested changes which should not take long to do. However, I think they are important to make to ensure that the relevant specialists will get the most out of this interesting study.

Best regards,

Michael Pittman
The University of Hong Kong

Referee expertise:

Referee #1: Molecular evolution, comparative genomics, phylogenetics, digestive enzymes

Referee #2: Evolution of digestive enzyme genes

Referee #3: Avian and flight origins

Reviewers' comments:

Reviewer #1 (Remarks to the Author):

The author has done a great deal of work to fully address my comments. I think the revised manuscript has been substantially improved. I have no more comments, and I am happy to recommend acceptance without further delay.

Reviewer #2 (Remarks to the Author):

The authors have carefully revised the manuscript according to the majority of the comments previously raised.

Reviewer #3 (Remarks to the Author):

Dear Dr. Wu (Hi Yonghua),

Thank you for making a range of changes to your manuscript, which have improved its quality. However, there are a number of revisions that I feel are necessary for me to support publication.

The use of *Aves* is debated, but most theropod palaeontologists use *Aves* to refer to the crown group. It is not wrong to use it as in O'Connor (2019) but you need to explicitly give the definition you are using in the introduction of your paper to avoid confusing readers that use *Aves* in different ways: I use *Aves* to mean...(sensu O'Connor 2019 which are based on XXXX). It is crucial to refer to the original definition (XXXX) and not just O'Connor (2019) as people won't know the definitions used in O'Connor (2019) from the reference but they will know if you list the original studies. Please see this chapter for a recent example of how to do this: Pittman, M., O'Connor, J., Field, D.J., Turner, A.H., Ma, W., Makovicky, P., and Xu, X. (2020). Pennaraptoran systematics. In Pennaraptoran theropod dinosaurs: past progress and new frontiers, M. Pittman and X. Xu, eds. (New York: Bulletin of the American Museum of Natural History), pp. 7-36. This is important to do --- otherwise different people will think you are talking about different things in your paper.

Reply: Done! We add the definition of Aves in our introduction. It reads like “Fossil evidence shows that ever since the origin of birds (Aves: defined herein as the clade including *Archaeopteryx* and modern birds, as proposed previously (Chiappe 1997; Mayr 2017a; O’Connor 2019)). Please see lines 43-44 for details.

The section "Main predators of ancestral birds—gliding non-avian maniraptorans" is still disjointed from the rest of the paper. You seem to use the paragraph to establish that non-avian paravians could have fed on avialans, and thus it sets up the following paragraph. If this is your intention, please help the reader by explicitly spelling out that this out is what the paragraph is doing -- now the first two lines talk about avialans being prey and the rest of the paragraph focuses on predatory characters of non-avian paravians so it is confusing to read.

Reply: Done! We replace the first two sentences as “Knowing the possible predators of ancestral birds is important to determine their potential predation risk.” Please see line 308 for details.

One important reference to add is Miller et al. 2020 which was just published in *Communications Biology* and is directly related to lines 240-241. This reference is also worth adding to the citation in lines 50-52 and adding *Confuciusornis* to the preceding sentence:

Miller, C. V., Pittman, M., Kaye, T. G., Wang, X., Bright, J. A., & Zheng, X. (2020). Disassociated rhamphotheca of fossil bird *Confuciusornis* informs early beak reconstruction, stress regime, and developmental patterns. *Communications Biology*, 3(1), 1-6.

Reply: Thank you for the recommendation of this paper. We cite the paper in our revised manuscript. Please see lines 49-51 and line 240 for details.

Figure 3 needs more work. Figure 3 includes Maniraptora but this is the wrong ancestor to talk about. It should be replaced with the more inclusive Pennaraptora node as this includes Oviraptorosauria which is the only other group of vane feathered non-paravian theropods. Many oviraptorosaurians are beaked and are thought to be herbivorous so this is highly relevant to your paper and should be marked in this figure (see Pittman & Xu 2020 Pennaraptoran theropod dinosaurs volume). Oviraptorosaurians are important to mark in Figure 3 and to mention -- at least briefly -- in the main text because non-paravian maniraptorans covers far too many groups that you do not mention in the paper and most of them aren't even covered in vane feathers. In that context, the Ma et al. (2017) is important to cite in your paper as an important paper on oviraptorosaurian diet:

Ma, W.S., Wang, J.Y., Pittman, M., Tan, Q.W., Tan, L., Guo, B., and Xu, X. (2017). Functional morphology of a giant toothless mandible from a bird-like dinosaur: Gigantoraptor and the evolution of the oviraptorosaurian jaw. *Scientific Reports* 7, 16247.)

Reply: Done! We agree with you that using Maniraptora as the ancestral node herein may be not correct, and Pennaraptora could be used. While considering that Fig.3 is mainly used to show the possible predation of ancestral birds by gliding non-avian paravians based on multiple lines of evidence, and in the revised manuscript we get the node of Maniraptora removed and only keep the clade of Paraves. The Pennaraptora and other members of non-paravian Maniraptora may be less relevant to our study since there is no relevant evidence indicating their possible predation on ancestral birds, and are not shown in our revised manuscript. Please see line 937 for the changes of Fig.3.

Whilst you want to keep Figure 3 simple, it is still misleading as it is. Saying a taxon with a fully edentulous jaw as having "partial tooth loss" is not correct. Please correct the figure as the reader

must currently read Wang et al. (2020) first to get the correct and full story of tooth which is not appropriate. You will not need to reproduce Wang et al. 2020 in full but the current Figure 3 needs to be improved or readers will think it contradicts recent work by a specialist on theropod tooth loss.

Reply: In Fig.3, we use black square, grey square and blank square respectively to represent complete dentition, partial tooth loss and complete tooth loss of a certain taxon based on previous papers (O'Connor 2019; Wang 2020). For instance, for *Confuciusornis*, which show complete tooth loss, we in Fig.3 use a blank square for it. Likewise, for Enantiornithes and Ornithuromorpha, both contain species with complete dentition, partial tooth loss, and complete tooth loss, respectively, and correspondingly we use three types of square (black, grey and blank) for each of them. Doing this does not contradict those published work (O'Connor 2019; Wang 2020). There may be some misunderstanding about the meaning of the symbols used, if any, and this explanation may help to understand about this.

Regarding alternative explanations for the results, the new paragraph is much better. For microbial fermentation, there is one bird to my knowledge (*Opisthocomus hoazin*) which does exhibit microbial fermentation of its food, and Alan Feduccia proposes it may have been present in enantiornithines (Feduccia 1999 page 150). I suggest looking at his evidence and deciding for yourself if this possibility is worth discussing further.

Feduccia, A. (1999). The origin and evolution of birds. 2nd edition. Yale University Press.

Reply: Thank you for the recommendation of the literature. Yes, some modern bird species including *Opisthocomus hoazin*, and possibly Enantiornithes, rely on microbial fermentation for nutrient utilization. While considering that microbial fermentation transfers dietary carbohydrates (e.g., cellulose) to volatile fatty acids and microbe proteins, and consequently leading to less selection enhancement of carbohydrate utilization. However, our study shows the marked selection enhancement of carbohydrate utilization of the common ancestor of living birds (CALB), suggesting that microbial fermentation is less important for CALB. Discussing the possible microbial fermentation of particular bird taxa may be less meaningful for our study.

This Miller and Pittman (2020) preprint (<https://www.essoar.org/doi/10.1002/essoar.10504068.2>) provides a much more comprehensive synthesis of avialan diet than O'Connor (2019), and thus should be cited in the citation lists of proposed bird diets as it is a DOI publication. The preprint is awaiting post-review acceptance in Biological Reviews which is expected in January 2020.

Reply: Done! Thank you for the recommendation of this paper. We cite this paper in our revised manuscript. Please see lines 50-51 for details.

I hope you can make these final requested changes which should not take long to do. However, I think they are important to make to ensure that the relevant specialists will get the most out of this interesting study.

Best regards,

Michael Pittman
The University of Hong Kong

REVIEWERS' COMMENTS:

Reviewer #3 (Remarks to the Author):

Hi Dr. Wu (Hi Yonghua!),

Thank you for making the changes requested. I am happy to recommend publication. Please thank me in the acknowledgements for my review of your manuscript.

Two last things about figure 3 that should be corrected.

1. 'gliding non-avian maniraptoran (predator)' is very strange as most non-avian maniraptorans cannot glide. Please replace with: gliding non-avian pennaraptorans (predator)

2. 'Paraves phylogeny with digestive system characteristics (gastric mill, crop, and tooth) and taxonomic definition (e.g., Aves) is based on one previous study (O'Connor 2019)' is a confusing sentence. Do you mean that the phylogeny used, mapping results and Aves definition are based on O'Connor 2019? If so, this should read: 'Paraves phylogeny with digestive system characteristics (gastric mill, crop, and tooth) and taxonomic definition (e.g., Aves) ARE based on a previous study (O'Connor 2019)'. Otherwise you need to provide a reference for the phylogeny.

Best regards,

Dr Michael Pittman (Mike)
The University of Hong Kong

REVIEWERS' COMMENTS:

Reviewer #3 (Remarks to the Author):

Hi Dr. Wu (Hi Yonghua!),

Thank you for making the changes requested. I am happy to recommend publication. Please thank me in the acknowledgements for my review of your manuscript.

Reply: Thank you very much for your review of our manuscript, and we thank you in the acknowledgement of our revised manuscript.

Two last things about figure 3 that should be corrected.

1. 'gliding non-avian maniraptoran (predator)' is very strange as most non-avian maniraptorans cannot glide. Please replace with: gliding non-avian pennaraptorans (predator)

Reply: Done!

2. 'Paraves phylogeny with digestive system characteristics (gastric mill, crop, and tooth) and taxonomic definition (e.g., Aves) is based on one previous study (O'Connor 2019)' is a confusing sentence. Do you mean that the phylogeny used, mapping results and Aves definition are based on O'Connor 2019? If so, this should read: 'Paraves phylogeny with digestive system characteristics (gastric mill, crop, and tooth) and taxonomic definition (e.g., Aves) ARE based on a previous study (O'Connor 2019)'. Otherwise you need to provide a reference for the phylogeny.

Reply: Done! We use "are" in our revised version. Thanks!